# An Ensemble of 3D U-Net Based Models for Segmentation of Kidney and Masses in CT Scans

Alex Golts[1], Daniel Khapun, Daniel Shats, Yoel Shoshan, and Flora Gilboa-Solomon

IBM Research
[1]`alex.golts@ibm.com`

**Abstract.** Automatic segmentation of renal tumors and surrounding anatomy in computed tomography (CT) scans is a promising tool in assisting radiologists and surgeons in their efforts to study these scans and improve the prospect of treating kidney cancer. In this paper we describe our approach to compete in the 2021 Kidney and Kidney Tumor Segmentation (KiTS21) challenge. Our approach is based on the successful 3D U-Net architecture with our added novelties including the use of transfer learning, an unsupervised regularized loss, custom postprocessing and multi-annotator ground truth that mimics the evaluation protocol.

**Keywords:** Semantic segmentation · medical imaging · 3D U-Net · kidney tumor

## 1   Introduction

Kidney cancer is among the 10 most frequently diagnosed cancer types [12] and among the 20 deadliest [19]. Surgery is the most common treatment option. Radiologists and surgeons diligently study the appearance of kidney tumors in CT imaging to facilitate optimal treatment prospects [4][11][15]. Automatic segmentation of kidney tumors and surrounding area is a promising tool in assisting them. It has already been proposed as a step in surgery planning [16], as well as enabled medical research around relating tumor morphology to surgical outcome [4][11].

The Kidney and Kidney Tumor Segmentation challenge of 2019 (KiTS19) [6] was the first to provide a public dataset with kidney tumor labels [7], boosting the available selection of segmentation algorithms specifically designed to segment kidney tumors. In KiTS19, 210 cases were given to participants for training. The kidneys and kidney tumors were annotated and the goal was to segment them accurately in 90 additional test cases.

Compared to KiTS19, the main changes in KiTS21 are:

1. The 90 test cases are now added to the training set which now includes a total of 300 cases. For the 2021 test, 100 additional cases are used.

2. A new segmentation class was added to the annotations, denoting cysts. Three Hierarchical Evaluation Classes (HECs) by which participants are evaluated are defined:
   1) **Kidney and Masses**: Kidney + Tumor + Cyst
   2) **Kidney mass**: Tumor + Cyst
   3) **Tumor**: Tumor only
3. A Surface Dice metric [13] was added for evaluation in addition to Sørensen-Dice.
4. Evaluation is performed against a random sample of aggregated segmentation maps which constitute plausible annotations in which different foreground class instances are labeled by different annotators.

Many of the successful algorithms for 3D segmentation in the medical domain are based on 3D variants of the popular U-Net architecture [3] [14]. Following its success and dominance as seen in the leading solutions in the KiTS19 challenge [6], we base our solution on the open source nnU-Net framework [10]. It offers automatic configuration of the different stages in a medical imaging segmentation task, including preprocessing, U-Net based network configuration and optional postprocessing.

Our proposed solution introduces several novelties:

- We employ a label sampling strategy during training to make use of the available multiple annotations, and to address the new evaluation protocol.
- We perform a form of transfer learning by initializing our network weights with those of a network pretrained on another public medical task.
- We augment the supervised training loss function with an unsupervised regularized term inspired by [5][17] which encourages similar prediction for neighboring voxels with similar intensity.
- We employ postprocessing which removes implausible tumor and cyst predictions that are disconnected from a kidney, as well as small kidney predicted fragments surrounded by another class.

The paper is structured as follows. In Sec. 2 we describe preprocessing and architectural details which were determined automatically by the nnU-Net framework [10]. Then in Sec. 3 we describe our unique decisions and contribution. These include our annotation sampling method, pretraining, proposed regularized loss, proposed postprocessing algorithm, and choice of models to use in a final ensemble. In Sec. 4 we provide experimental results. Finally, in Sec. 5 we conclude.

## 2   nnU-Net determined details

### 2.1   3D U-Net Network architecture

The U-Net [14] is an encoder-decoder network. The decoder receives semantic information from the end of the encoder (bottom of the "U") and combines it through skip connections with higher resolution features from different layers

of the encoder. In our 3D U-Net variant all convolution kernels are $3 \times 3 \times 3$. Each block in the encoder consists of a sequence of Conv-InstanceNorm[18]-LeakyReLU operations repeated twice. In the encoder, one of these Conv operations has a stride of 2 to facilitate downsampling. In total there are 5 downsampling operations. In the decoder, the same number of upsampling operations is done via transposed convolutions.

In most of our experiments, we apply the above architecture as a single stage network which gets a preprocessed image patch (Sec. 2.3 as input and outputs a final segmentation map. However, we also experimented with a two-stage architecture described next.

## 2.2    3D U-Net Cascade Network architecture

The 3D U-Net cascade is another network type offered in the nnU-Net framework. It serves the purpose of increasing the spatial contextual information that the network sees, while keeping a feasible input patch size with regards to the GPU memory. This can be achieved by applying a 3D U-Net on downsampled, lower resolution input data. However, this comes at a cost of reduced detail in the generated segmentations. Therefore, a second stage is performed in which another 3D U-Net is applied on high resolution input data. In the second stage, the high resolution input is augmented with extra channels that contain the one-hot encoded segmentation maps generated by the "low resolution" 3D U-Net from the first stage. These maps are first upsampled to the higher resolution input data size. Fig. 1 depicts the 3D U-Net cascade in high level. In our case,

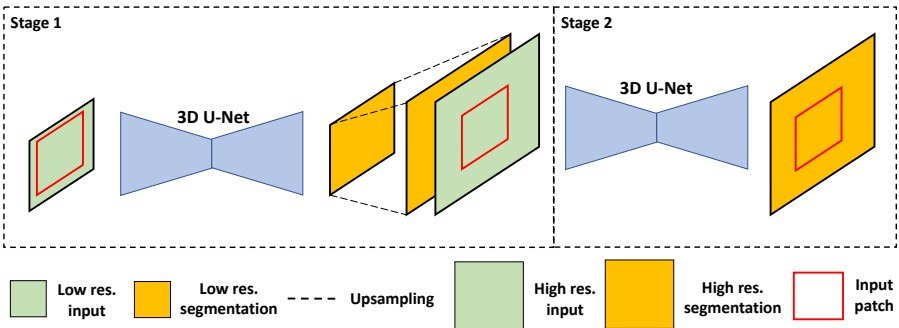

**Fig. 1.** The 3D U-Net cascade model. In the first stage, low resolution input enters a 3D U-Net. The patch size covers a large portion of the image, facilitating rich contextual information. Then, the low resolution segmentation is upsampled and concatenated with the high resolution input. In the second stage, the input patch covers a smaller portion of the input, but global segmentation information is already available. Then, a second 3D U-Net is applied and refines the segmentations, obtaining them in high resolution.

nnU-Net determined the 1st stage 3D U-Net to be of the same structure as the 2nd stage network, as detailed in Sec. 2.1.

### 2.3   Preprocessing

The median voxel spacing in the original training data is $0.78 \times 0.78 \times 3.0$ mm. The median volume shape is $512 \times 512 \times 109$ voxels.

We clip each case's intensity values to the 0.5 and 99.5 percentiles of the intensity values in the foreground regions across the training set, which correspond to the range [-62, 310]. Then, we subtract the mean and divide by the standard deviation of the intensities in the foreground regions, which correspond to 104.9 and 75.3, respectively.

During training, patches with shape $128 \times 128 \times 128$ are sampled and input to the network. To increase training stability, the patch sampling enforces that more than a third of the samples in a batch contain at least one randomly chosen foreground class.

**2.3.1   Low resolution**  In the 3D U-Net cascade model, the 1st stage, low resolution network operates on input data resampled to a common spacing of $1.99 \times 1.99 \times 1.99$ mm. This results in median volume shape of $201 \times 201 \times 207$ voxels for the training cases.

**2.3.2   High resolution**  In the single stage 3D U-Net models, and the 2nd stage of the 3D U-Net cascade, the network operates on input data resampled to a common spacing of $0.78 \times 0.78 \times 0.78$ mm. This results in median volume shape of $512 \times 512 \times 528$ voxels for the training cases.

### 2.4   Training details

Beside our proposed regularized loss term (Sec. 3.4) all our models are trained with a combination of dice and cross entropy loss [10]. The loss is applied at the 5 different resolution levels in the decoder.

Training is done on a single Tesla V100 GPU. The models train for 1000 epochs with each epoch consisting of 250 iterations with a batch size of 2. For the single stage, full resolution 3D U-Net model, training takes about 48 hours.

We use an SGD optimizer with Nesterov momentum of 0.99, and learning rate which decays in each epoch according to $\text{lr} = 0.01 \left(1 - \frac{\text{epoch}}{1000}\right)^{0.9}$.

## 3   Method

Our solution is based on 3D U-Nets with several novel additions. We found that a 2D U-Net, although faster to train, results in significantly inferior performance on the tumor and cyst classes. For the kidney class, performance is on par

with a 3D U-Net. We found at a late stage in the competition that the 3D U-Net cascade model performs better than the single stage 3D U-Net. Therefore, most of our unique contribution was demonstrated using the single stage 3D U-Net model. For the final submission, we ensemble three such models with one cascade model. Next, we describe the training and validation data followed by our unique decisions and contribution which include the annotation sampling method, optional pretraining, proposed regularized loss, proposed postprocessing algorithm and choice of models for ensemble.

### 3.1   Training and Validation Data

We use only the official KiTS21 300 training cases for our submission. The only way in which we *indirectly* use other publicly available data in some of our experiments, is by initializing network weights with those of a model pretrained on the Liver Tumor Segmentation (LiTS) database [1].

We train our models on 5 cross validation splits of 240 training cases and the remaining 60 used for validation. The splits are randomly decided by nnU-Net[10]. In Sec. 4 we report average Dice and Surface Dice scores over the cross validation splits per HEC, as well as global averages across the HECs. The evaluation metrics are computed usign the competition's official evaluation function.

### 3.2   Pretraining

In Sec. 4.1.2 we show the effect of initializing the network weights from a pre-trained model trained on LiTS[1]. This is in contrast to other experiments in which the network weights were initialized randomly. We note that the specific pretrained model we used is available for download under the nnU-Net framework and fits the 3D U-Net network structure determined for our data without any modification. This allowed us a simple way of testing a form of transfer learning for our task.

### 3.3   Annotations

In KiTS21, each kidney/cyst/tumor instance has been annotated multiple times by different annotators. The competition organizers provided a script for generating a (seeded) random plausible aggregated segmentation maps for each case. There are between 6 and 15 such maps per case, depending on the case's number of annotators and class instances. During evaluation, the competition metrics for each case are computed and averaged against all the case's sampled plausible maps.

To resemble the official evaluation protocol, we wanted models to see different plausible annotations during training. We achieve this by choosing for each slice within a case (volume), a random plausible annotation map out of the 6 to 15 options that are available after running the sampling script. We use this random choice as the set of training annotations. In Sec. 4.1.1 we show the effect on performance of using different random seeds for this training annotation selection procedure.

### 3.4    Regularized loss

Previous works on weakly supervised segmentation tasks have proposed to add to the "standard" loss term which makes use of the existing supervised label seeds, another term which is unsupervised. It does not require labels as input, but only the original signal (raw volume in our case), and the network prediction [5][17]. Intuitively, this loss term should encourage the predictions to follow a desired behaviour, such as to be smooth in some sense. In semantic segmentation we might want the loss to penalize contradicting predictions for neighboring voxels which are similar in their intensity. We experiment with a regularized loss proposed in [5] which can be thought of as a special case of the Potts model [2]. We denote the regularized loss $L_{\mathrm{reg}}$. Then, the total training loss for training our 3D U-Net becomes

$$L_{\mathrm{total}} = L_{\mathrm{dice}} + L_{CE} + \lambda L_{\mathrm{reg}} \ , \tag{1}$$

where $L_{\mathrm{dice}}$ and $L_{CE}$ denote the dice and cross-entropy losses, respectively, and $\lambda$ is a hyperparameter.

**3.4.1    Image loss** In one of our attempts, we used the regularized loss proposed in [5] for a 2D image, and applied it to each volume slice. This loss affects each pixel through its four upper, lower, right and left neighbors. Let $\mathbf{I}$ be an image, $i$ and $j$ denote two neighbors, $\varepsilon$ is the pixel's mentioned 4-neighborhood, $\mathbf{p}^c$ is the predicted segmentation softmax score for class $c$. The loss is then given by

$$L_{\mathrm{reg}}(\mathbf{I}, \mathbf{p}) = \sum_c \sum_{(i,j) \in \varepsilon} w_{ij} \left( p_i^c - p_j^c \right)^2 \ , \tag{2}$$

where

$$w_{ij} = e^{-\beta (I_i - I_j)^2} \ . \tag{3}$$

**3.4.2    Volume loss** Here we generalize the regularized loss to also account for 2 additional forward and backward neighbors from the adjacent slices. Eqs. 2-3 remain the same, but $\varepsilon$ now contains 6 neighbors instead of four.

### 3.5    Postprocessing

We applied a postprocessing algorithm on the segmentation results, that removes rarely occuring implausible findings. The algorithm consists of two parts

1. **Tumor and cyst finding positioned outside of kidney findings are removed.**
   We compute a slightly dilated mask of 3D connected components of voxels classified as tumor or cyst. For each dilated connected component, if it has no intersection with at least one kidney classified voxel, then we change the classification of the corresponding tumor of cyst finding to "background".

2. **Small kidney fragments surrounded by another class are removed.**
   We compute 3D connected components of voxels classified as kidney. We select all components smaller in volume than the 3rd largest. Then, we change the classification of those smaller components, to that of the majority of the voxels in its slightly dilated surrounding.

### 3.6   Final submission

For the final submission, we use an ensemble of four models, three single stage high resolution 3D U-Nets and one 3D U-Net cascade. Each of the four models is an ensemble on its own, of its 5 trained cross validation folds. Our postprocessing (Sec. 3.5) was applied on the final segmentations output after the ensemble. The following is a description of each model in our final ensemble:

1. **3D U-Net** trained with the regularized loss from Sec. 3.4.1.
2. **3D U-Net** for which training was initialized with a model pretrained on LiTS. (Sec. 3.2).
3. **3D U-Net** trained with a different random seed for the training annotation generation process (Sec. 3.3) than the other three models in the ensemble.
4. **3D U-Net cascade** in which training of the 1st stage, low resolution network was initialized with a model pretrained on LiTS (Sec. 3.2).

## 4   Results

In all experiments we use the official evaluation code which calculates Dice and Surface Dice metrics averaged across sampled plausible annotation maps. The results we show here are all average scores across 5 cross validation splits. For brevity we denote in the tables in this section the per HEC dice scores as $D1, D2, D3$ for the "Kidney and Masses", "Kidney mass" and "Tumor" HECs, respectively, and their mean is denoted MD. Similarly, surface dice scores are denoted $SD1, SD2, SD3$, and their mean MSD.

### 4.1   Single stage, high resolution 3D U-Net

The following experiments were made based on the single stage 3D U-Net model (Sec. 2.1).

**4.1.1   Random annotations** In Tab. 1 we show how our 3D U-Net trained with random annotation procedure as described in Sec. 3.3 performed against a baseline that we trained which used aggregated maps according to a majority vote (MAJ). We also show results of the baseline published in [9].

We see that the random annotation procedure unfortunately has no significant effect on performance. We still decided to add the model with seed 3 to our final model ensemble. In all the next experiments, we use our random annotation procedure with seed 1.

| Model | D1 | D2 | D3 | SD1 | SD2 | SD3 | MD | MSD |
|---|---|---|---|---|---|---|---|---|
| baseline ([9]) | 0.9666 | 0.8618 | 0.8493 | 0.9336 | 0.7532 | 0.7371 | 0.8926 | 0.8080 |
| baseline (our) | 0.9660 | 0.8589 | 0.8444 | 0.9334 | 0.7506 | 0.7320 | 0.8897 | 0.8053 |
| seed 1 | 0.9662 | 0.8583 | 0.8449 | 0.9335 | 0.7513 | 0.7330 | 0.8898 | 0.8059 |
| seed 2 | 0.9655 | 0.8581 | 0.8419 | 0.9324 | 0.7500 | 0.7297 | 0.8885 | 0.8040 |
| seed 3 | 0.9668 | 0.8645 | 0.8478 | 0.9347 | 0.7567 | 0.7356 | **0.8930** | **0.8090** |

**Table 1.** Our random annotation procedure vs. baseline MAJ annotations

**4.1.2    Pretraining** In Tab. 2 we show the effect of transfer learning, namely initializing our 3D U-Net model from weights of a model pretrained on LiTS.

| Model | D1 | D2 | D3 | SD1 | SD2 | SD3 | MD | MSD |
|---|---|---|---|---|---|---|---|---|
| baseline ([9]) | 0.9666 | 0.8618 | 0.8493 | 0.9336 | 0.7532 | 0.7371 | 0.8926 | 0.8080 |
| baseline (our) | 0.9660 | 0.8589 | 0.8444 | 0.9334 | 0.7506 | 0.7320 | 0.8897 | 0.8053 |
| with pretraining | 0.9674 | 0.8651 | 0.8518 | 0.9346 | 0.7563 | 0.7396 | **0.8948** | **0.8102** |

**Table 2.** Transfer learning from LiTS

We see some improvement, therefore we add the model with initialization from a model pretrained on LiTS to our final model ensemble.

**4.1.3    Regularized loss** In Tab. 3 we show the effect adding regularized loss, as described in Sec. 3.4. Following limited hyperparameter search, we use $\beta = 10$ for the image loss, $\beta = 5$ for the volume loss, and $\lambda = 1$ for both versions. We

| Model | D1 | D2 | D3 | SD1 | SD2 | SD3 | MD | MSD |
|---|---|---|---|---|---|---|---|---|
| baseline ([9]) | 0.9666 | 0.8618 | 0.8493 | 0.9336 | 0.7532 | 0.7371 | **0.8926** | **0.8080** |
| baseline (our) | 0.9660 | 0.8589 | 0.8444 | 0.9334 | 0.7506 | 0.7320 | 0.8897 | 0.8053 |
| image loss | 0.9659 | 0.8615 | 0.8493 | 0.9341 | 0.7528 | 0.7370 | 0.8922 | **0.8080** |
| volume loss | 0.9663 | 0.8609 | 0.8482 | 0.9336 | 0.7512 | 0.7337 | 0.8918 | 0.8062 |

**Table 3.** Performance with regularized loss

add the model with image regularized loss to our final model ensemble, as it showed slight improvement over at least *our* baseline. Our experiments showed that hyperparameter tuning for $\beta$ and $\lambda$ are important for the regularized loss. This could be one of the reasons we did not manage to better optimize the volumetric version of the loss within our time and resources constraints. It is also why we did not choose to employ this loss in conjuction with other steps like initializing with a pretrained model, or the next experiment with the 3D U-Net cascade model. We suspected that separate hyperparameter tuning could need to be performed for each scenario.

### 4.2   3D U-Net cascade

In Tab. 4 we show the performance of our trained 3D U-Net cascade model (Sec. 2.2 compared to the baseline published in [9] (for the same model type).

| Model | D1 | D2 | D3 | SD1 | SD2 | SD3 | MD | MSD |
|---|---|---|---|---|---|---|---|---|
| cascade baseline ([9]) | 0.9747 | 0.8799 | 0.8491 | 0.9453 | 0.7714 | 0.7393 | 0.9012 | 0.8187 |
| our cascade | 0.9747 | 0.8810 | 0.8583 | 0.9459 | 0.7709 | 0.7461 | **0.9046** | **0.8210** |

**Table 4.** Performance of our 3D U-Net cascade model

We see some improvement for our cascade model over the published baseline. This could be owed to our random annotation procedure (Sec. 3.3) and our initialization of the 1st stage low resolution 3D U-Net with a model pretrained on LiTS (Sec. 3.2).

### 4.3   Model ensemble

In Tab. 5 we show the effect ensembling four models as described in Sec. 3.6. We also show for comparison an ensemble of only the three single-stage 3D U-Net models, as well as best single models (without ensemble), out of both single-stage and cascade. Again, metrics are averaged across all 5 cross validation splits.

| Model | D1 | D2 | D3 | SD1 | SD2 | SD3 | MD | MSD |
|---|---|---|---|---|---|---|---|---|
| best model (cascade) | 0.9747 | 0.8810 | 0.8583 | 0.9459 | 0.7709 | 0.7461 | **0.9046** | 0.8210 |
| best 1-stage model | 0.9674 | 0.8651 | 0.8518 | 0.9346 | 0.7563 | 0.7396 | 0.8948 | 0.8102 |
| ensemble of 1-stage models | 0.9674 | 0.8667 | 0.8535 | 0.9363 | 0.7610 | 0.7442 | 0.8959 | 0.8138 |
| final ensemble | 0.9702 | 0.8751 | 0.8597 | 0.9400 | 0.7709 | 0.7525 | 0.9017 | **0.8211** |

**Table 5.** Model ensemble

We see that the cascade model, which outperformed all the others, is alone better than the final ensemble in terms of the average Dice score. But we do see a slight improvement in average Surface Dice, and also in the tumor class metrics, which are arguably the most critical in practice (and also the tumor Dice is used as a tiebreaker in KiTS21). We also see that ensembling the three single-stage 3D U-Net models improves over the best model among them. Therefore we decided to ensemble all four models in our final submission.

### 4.4   Postprocessing

In Tab. 6 we show the result of applying our proposed postprocessing algorithm (Sec. 3.5) to the segmentation results of our final ensemble. Again, metrics are averaged across all 5 cross validation splits. We see improvement across all met-

| Model | D1 | D2 | D3 | SD1 | SD2 | SD3 | MD | MSD |
|---|---|---|---|---|---|---|---|---|
| without postprocessing | 0.9702 | 0.8751 | 0.8597 | 0.9400 | 0.7709 | 0.7525 | 0.9017 | 0.8211 |
| with postprocessing | | 0.9715 | 0.8790 | 0.8638 | 0.9415 | 0.7751 | 0.7569 | **0.9047** | **0.8245** |

**Table 6.** Results with postprocessing applied to our final ensemble segmentations

rics.

Fig. 4.4 shows an example from case 16 in the database, predicted using our model ensemble. Specifically, for the demonstration to be fair, the ensembled model of cross validation fold 0, in which case 16 was part of the validation set. In the first row, we see slice 60, which contains kidney (red) and tumor (green) findings. In the second row, slice 105, in which a false tumor finding was predicted, and successfully removed after applying our postprocessing algorithm, since it has no contact with a kidney prediction.

| **original image** | **prediction without postprocessing** | **prediction with postprocessing** | **ground truth** |

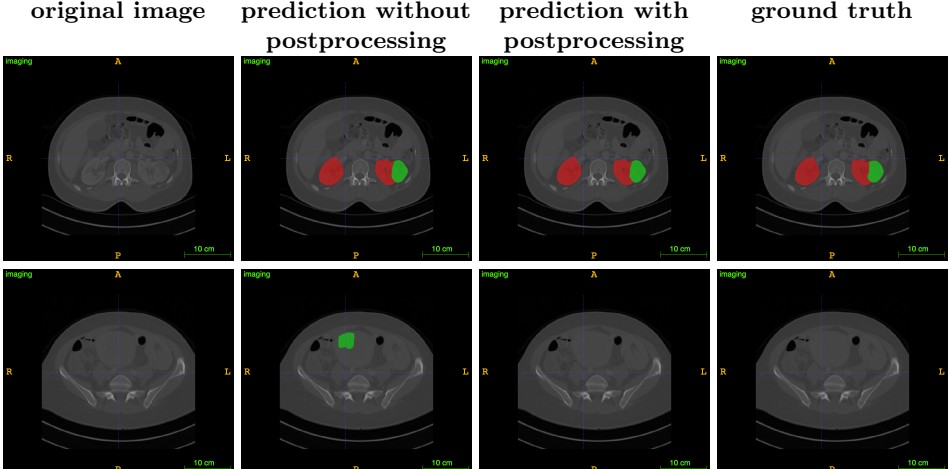

**Table 7.** Example predictions for case 16. Top row: slice 60 which contains kidney (red) and tumor (green) findings. Bottom row: slice 105 which contains a false tumor prediction, successfully removed by our postprocessing algorithm.

## 5    Discussion and Conclusion

We presented results of our 3D U-Net based approach to solving the KiTS21 challenge. We managed to demonstrate minor improvements over published baselines based on a single-stage 3D U-Net, as well as a two-stage 3D U-Net cascade. Improvements are owed, to varying degrees, to our following contributions: a method for utilizing multiple annotations during training, weight initialization from a model pretrained on a different task, an unsupervised term added to

the loss function that encourages smoothness in the segmentation predictions, ensembling of multiple models, and a proposed postprocessing algorithm.

The participation in the challenge leaves us with quite a few interesting directions for future research. We now realize that better performance could be reached if we applied some of our contributions to the better performing 3D U-Net cascade model, rather than the single-stage 3D U-Net. The regularized loss could benefit from more thorough hyperparameter tuning as well as further generalization to use more neighboring voxels. Additionally, more recent network architectures for semantic segmentation are worth exploring. Outside the scope of this particular challenge, it is worth investigating the trade-off between accuracy and runtime in medical imaging segmentation, for example as is evident when comparing 2D and 3D U-Net architectures.

As we look to experiment with different network architectures, or work on extending the idea of regularized losses for medical imaging segmentation, we may opt for open source frameworks designed for flexible and efficient research in the medical imaging domain. One such promising framework is the recently released FuseMedML library [8].

## Acknowledgment

We thank the KiTS competition organizers, data providers and annotators for their great effort in advancing the science around kidney cancer, including improving algorithms for automatic segmentation. We further thank the creator and contributors to the nnU-Net framework, an excellent and generally applicable baseline for medical segmentation tasks.

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
