# OpenReview forum: "An Ensemble of 3D U-Net Based Models for Segmentation of Kidney and Masses in CT Scans"
_MICCAI.org/2021/Challenge/KiTS — Submitted to KiTS21 Challenge_

### Official Review · Reviewer_d2m2 · 2021-08-30

**Rating:** 9

**Review:**

This is an excellent paper. The methods are described in great detail and results are presented from several experimental trials, along with an explanation of why the final approach was chosen. The paper could benefit from a figure or two but the clarity of writing makes up for the lack of figures. The authors should make sure to add the final results once they are known.

---

### Official Review · Reviewer_BhDo · 2021-08-30

**Rating:** 10

**Review:**

### Overall

- Great job with this paper. Please make sure to add the final results once they are known.

### Introduction

- Looks good

### Methods

- What algorithm did you use to resample the imaging and labels?
- It might be nice to add a figure to visually summarize your approach

### Results

- Excellent job with including results for multiple experiments in detailed tables
- It might be nice to add an example prediction vs ground truth

### Discussion and Conclusion

- Looks good

---

### Decision · Program_Chairs · 2021-08-30

**Decision:**

Minor Revisions

**Comment:**

Please address the reviewer comments and resubmit